# Selection of Optimal Palmer Predictors for Increasing the Predictability of the Danube Discharge: New Findings Based on Information Theory and Partial Wavelet Coherence Analysis

**DOI:** 10.3390/e24101375

**Published:** 2022-09-27

**Authors:** Ileana Mares, Constantin Mares, Venera Dobrica, Crisan Demetrescu

**Affiliations:** Institute of Geodynamics of the Romanian Academy, R-020032 Bucharest, Romania

**Keywords:** mutual information, nonlinear correlation, redundancy–synergy index, partial wavelet coherence, Palmer index, Danube discharge

## Abstract

The purpose of this study was to obtain synergistic information and details in the time–frequency domain of the relationships between the Palmer drought indices in the upper and middle Danube River basin and the discharge (Q) in the lower basin. Four indices were considered: the Palmer drought severity index (PDSI), Palmer hydrological drought index (PHDI), weighted PDSI (WPLM) and Palmer Z-index (ZIND). These indices were quantified through the first principal component (PC1) analysis of empirical orthogonal function (EOF) decomposition, which was obtained from hydro-meteorological parameters at 15 stations located along the Danube River basin. The influences of these indices on the Danube discharge were tested, both simultaneously and with certain lags, via linear and nonlinear methods applying the elements of information theory. Linear connections were generally obtained for synchronous links in the same season, and nonlinear ones for the predictors considered with certain lags (in advance) compared to the discharge predictand. The redundancy–synergy index was also considered to eliminate redundant predictors. Few cases were obtained in which all four predictors could be considered together to establish a significant information base for the discharge evolution. In the fall season, nonstationarity was tested through wavelet analysis applied for the multivariate case, using partial wavelet coherence (*pwc*). The results differed, depending on the predictor kept in *pwc*, and on those excluded.

## 1. Introduction

Many factors contribute to the evolution of a natural geophysical phenomenon. In a complex interdependence, there may be associated factors that mask the determining causes of the respective geophysical phenomena. The determination of these dependencies or interdependencies is the basis of a robust objective prediction of the geophysical phenomenon.

Traditionally, the detection of the prescribed multivariable signal was performed using a robust fingerprint technique, maximizing the signal-to-noise ratio of the associated detector (Hasselmann, 1993) [1].

Relatively recent investigations [2,3,4,5,6,7,8,9,10,11,12,13,14,15] provide us with a series of analysis procedures that elucidate the intrinsic transition between the states of a natural phenomenon subject to disturbing causes. Therefore, we were able to decipher them to an extent.

The aim of the present study is to find an application that outlines synergistic information modules within a complex system, more specifically, finding the magnitude of the specific contribution of useful synergistic information between the set of X predictors and a Y predictand.

This can be carried out using the framework of information theory, which was initially outlined by Shannon (1948) [16], who introduced informational entropy, as a measure of indeterminacy for a priori events, or as a measure of information for events occurring posterior to the experiment, as shown by Guiasu (1977) [17].

The first part of this study assumes a special extension of a hydrological predictability framework based on information contained in indices that also capture aspects of hydro-climatic variability to answer questions that have not been fully elucidated regarding spatiotemporal connectivity. Therefore, the main purpose of this work is to obtain details regarding the information contained within the Palmer drought indices, which could be found in the behavior of the Danube discharge.

The methods presented here complement previous studies conducted in different geographical areas [11,18,19,20] and are generally useful for implicitly understanding aspects of short-term climate variability and its hydrological impact.

In general, the predictability of the respective hydrological regime depends on the characteristics of the area for which the information content of the predictors is analyzed [11]. However, in an analysis, we start with the simplest task, which is investigating the type of links between the predictor and predictand: linear connections with measures, such as simple correlation coefficients involving Gaussian distributions, as well as nonlinear links with non-Gaussian distributions, based on Shannon entropy.

A global correlative measure was described by Watanabe (1960) [21] and Timme et al. (2014) [22] as “total correlation (TC)”. Ball et al. (2017) [5], referencing Matsuda (2000) [23], shows that TC is a more attractive measure because the joint entropy of multiple variables “is guaranteed to be non-negative since the joint entropy of multiple variables must be less than or equal to the sum of the entropies of the individual variables”. This property is also demonstrated in Relation (P10) from Guiasu (1977) [17].

The most appropriate measure of the complexity of the interaction is given by what is called Normalized Mutual Information (NMI) (after Ball et al., 2017) [5]. NMI is a scalar measure of global mutual information in a multivariate network of interactions. NMI reduces the effect of an additional artificial increase in the total correlation. However, is it sufficient to use only the global NMI measure to reduce the possible effect of a further artificial increase in the TC correlative link, and thus not produce a “spurious correlation”? NMI does not have this capability, as found in several investigations [19,22], because it has been found that there is redundant information on predictors that mask the true connection with the predictand. The measure of the relationship between synergy and redundancy has been carried out by several investigators [22,24,25].

In the present paper, we apply the redundancy–synergy index (RSI) variant (Timme et al., 2014) [22]. The importance of this measure is described herein.

In the second part of the paper, the details of the interactions are then determined by applying wavelet transform to a multivariable case. In this way, considering the consistency with the nonstationary and nonlinear structures of the time series, the advantage of the principle of overlap [26,27] is ensured, as well as the limits suggested by the principle of uncertainty regarding the time–frequency details [28,29,30]. The advantages of using wavelet transform are set out in many papers [30,31,32,33,34,35,36,37,38,39].

In the case of applying wavelet transform to more than two variables, due to the existing interaction between the contributing variables, a specific bivariate relationship can be clearly explained only by determining the role of other contributing variables. This can be facilitated by performing a partial wavelet coherence analysis (*pwc*) by excluding the effect of other predictive variables.

Thus, applications of wavelet transform to multiple variables are found in many investigations [3,4,12,15,40,41,42,43].

We believe that this study will facilitate the inclusion (according to the signals at different spatiotemporal scales) of hydrological cycles in climate prediction/simulation models according to certain requirements (Gallegati, 2022) [29] to capture the variability and interaction of different components of the climate system. To date, the in-depth extraction of information from predictors with important nonlinear/non-stationary links for the discharge of a river such as the Danube on natural basins of attraction (seasons specific to middle latitudes) based on information theory has not been carried out. Additionally, the more-or-less natural mechanism of climate change can be untangled, thanks to the progress made in recent years through wavelet analysis. Through wavelet analysis, it is possible to immediately detect, from a set of predictors, important signals for river discharge prediction. The importance of this for economic/social evolution was seen even this year (2022) with the hydrological extremes in the Danube River basin.

The paper is structured as follows: Section 2 (Materials and Methods) contains two paragraphs. The first paragraph (2.1) describes the data used as predictors, i.e., Palmer drought indices and the predictand variable consisting of the Danube discharge at the Orsova station located in the lower Danube River basin. In Section 2.2., the applied methods are presented, namely, those that are based on the mutual information applied for the calculation of the total correlation, as well as the redundancy–synergy ratio in the multivariate case.

The procedures for applying wavelet transform in the multivariate case are also briefly described, especially for the application of partial wavelet coherence (*pwc*). The results and discussions are described in Section 3. It has three parts: in 3.1., the results are presented after testing the nonlinearities between the predictors and predictand; in 3.2., the total correlations and the ratio between synergy and redundancy are presented and discussed; and in 3.3., the discussion focuses on the results obtained in partial wavelet coherence. Section 4 presents the conclusions and the next steps that should be followed in this investigation.

This study is based on time series from the twentieth century (1901–2000), and investigations were performed separately for each of the four seasons. This time interval and the seasonal averages were considered in order to be able to compare the results obtained here with those obtained in our previous investigations [18,19,20,44].

## 2. Materials and Methods

### 2.1. Data

As predictors for the Danube discharge, we analyzed four versions of the Palmer index. Three of the Palmer indices used in the present study, namely, the Palmer drought severity index (PDSI), the Palmer hydrological drought index (PHDI) and the Palmer Z index (ZIND), appear in the initial model proposed by Palmer (1965) [45]. The fourth index used is the modified Palmer drought severity index (WPLM), which, as mentioned by Vicente—Serrano et al. (2012) [46], was proposed by the National Weather Service Climate Analysis Center for operational meteorological purposes (Heddinghaus and Sabol 1991) [47], modifying the original rules of accumulation during wet and dry spells. Although the term ‘drought’ appears in the name of the Palmer drought index, it still highlights both wetness (positive values) and dryness (negative values).

Monthly values of air temperature, the total of monthly precipitation and the corresponding Available Water Capacity (AWC) values for the 15 stations (Figure 1) situated in the Danube upper and middle basin, for the period 1901–2000, were considered in order to calculate these Palmer indices. The selection of stations was performed according to their position in the Danube River basin or the tributaries of the river.

In the present study, the analysis was extended over the period 1901–2000. The values of monthly precipitation and temperature were obtained from the CRU TS3.10 Dataset. Details are given in Harris et al. (2014) [48].

Datasets were generated as high-resolution (0.5 × 0.5°) grids using the Climatic Research Unit (CRU), and we selected (with the respective coordinators) the option ‘half grid points’ for each station.

Details on data quality can be found in the work of Harris et al. (2014) [48]. We compared the interpolated data with the observational data available from the stations obtained by the European Climate Assessment & Dataset project (ECA&D) (Klein Tank et al. (2002) [49]) for the available and common periods, and the differences are insignificant. Details regarding the data quality from the ECA&D can be found in the work of Klein Tank et al. (2002) [49] and Wijngaard et al. (2003) [50]. The characteristics of the 15 considered stations are indicated in Table 1, and their position in relation to the hydrological station Orsova is described in Figure 1.

The AWC values used in the present study were extracted from the Harmonized World Soil Database (FAO/IIASA/ISRIC/ISSCAS/JRC, 2012).

For each of the 15 stations for the period 1901–2000, the original Palmer indices were calculated using a routine performed by Nathan Wells (2003) [51] from the National Agricultural Decision Support System. The input variables in Wells’ routine were the temperatures, precipitation and a parameter.

The parameter file contains two numbers. The first number is AWC. The second number, by default, should be the latitude of the station provided in Table 1. A brief review of Palmer’s procedure, along with the respective equations, is performed in the work of Wells et al. (2004) [52].

In the same publication, the authors describe a calibration procedure, i.e., introduce a self-calibrating Palmer drought severity index (SC-PDSI). The authors of this study compared, in the work of Mares et al. (2016) [44], the influence of the four original Palmer indices on the discharge of the Danube River at Orsova with the influence of the self-calibrating ones. Compared to the self-calibrating indices, the original ones for seasons are more suitable from a statistical point of view, regarding the influence on Danube discharge in the lower basin. The comparison was made both for the first principal component (PC1) of the EOF for each index (both original and self-calibrating) and for the PC1 of the development in the Multivariate Empirical Orthogonal Functions of the four indices (PC1-MEOF). For this reason, in the present study, we focused only on the original Palmer indices.

In the current investigation, for each of the four Palmer indices, we developed, in orthogonal empirical functions (EOFs), the time series from the 15 stations and considered only the first principal component (PC1), although this is not mentioned in the remainder of the paper.

Although there are several types of indices that highlight the dry and wet periods [19,46,53], in our study, we used only the four Palmer indices, which were of the same type. Therefore, we could conduct a comparison between the influence of factors captured by each of them and the influence of all four indices on the Danube discharge.

The predictand variable is the Danube discharge (Q) at Orsova. This station, although located in the lower basin of the Danube River (Figure 1), represents an integrator of the precipitation from the upper and middle basin, which is located at the entrance of the Danube River into the lower basin. Data were provided by the National Institute of Hydrology and Water Management, Bucharest, Romania. In the Appendix A, the discharge of the Danube River to Orsova is abbreviated to Q_ORS.

### 2.2. Methods

#### 2.2.1. Elements of Information Theory

The information content of the 4 predictors (Palmer indices) was obtained first by applying the elements of information theory; then, to acquire details of the frequency–time domain, elements of the multivariate wavelet transform were applied, such as multiple wavelet coherence (*mwc*) and partial wavelet coherence (*pwc*).

The concept of Shannon’s information entropy (1948) [16], which is very widely used in hydrology [2,9,54,55,56,57,58,59,60,61,62,63], was also applied in the present work.

For a discrete time series, information entropy in accordance with Shannon’s definition is
(1)$$H(X)=-\sum_{x€X}p(x)\;\log_{b}\;p(x)$$

The entropy is measured in bits if *b =* 2, and in nats if *b = e*. In the present study, we used *b =* 2.

By means of joint entropy
(2)$$H(X,Y)=-\sum_{x€X}\sum_{y€Y}p(x,y)\;\;\log\;p(x,y)$$
we can estimate mutual information (MI)
(3)MI(X,Y)=H(X)+H(Y)−H(X,Y)

Mutual information indicates the amount of information shared between the two time series.

Using MI, we can find a nonlinear measure between the analyzed variables, such as the nonlinear correlation coefficient (NLR), which is defined as follows:(4)$$NLR=\sqrt{\left[1-\exp(-2MI)\right]}$$

The NLR variation range is the same as the linear correlation coefficient but has the advantage of ignoring the statistical distribution of the variables analyzed or the nature of their relationship. Many studies detail the performance of this nonlinear measure in comparison with classical linear correlation (Khan et al., 2006, Vu et al., 2018) [56,63].

In the present study, we compared NLR values calculated according to Equation (4) with the Pearson linear correlation coefficient, between each of the four predictors (Palmer indices) and the predictand (Danube discharge at Orsova).

The four indices were abbreviated as follows: all Palmer indices are abbreviated as PI; I1, I2, I3 and I4 correspond to PDSI, PHDI, WPLM and ZIND, respectively. All indices represent the first principal component (PC1) of the development in EOF, but for simplicity, this was not specified.

All the investigations were performed for seasonal averages, considered in the following order and abbreviated as follows: winter (WIN = s1), spring (SPR = s2), summer (SUM = s3) and fall (FALL = s4). The correlations were achieved with lags from 0 to 3 seasons in comparison with the Danube discharge (Q) to maintain the total length of the series of 100 years (1901–2000). The combinations of the 4 predictors considered simultaneously and with delays from 1 to 3 seasons resulted in 10 situations for each of the predictors, which are presented schematically in Appendix A. The nonlinear correlation coefficients (NLR) and the linear ones (R), together with their confidence levels (CL), are presented in Appendix A.

Since through NLR and R, we only obtained information transmitted to predictand Q separately by each of the four predictors considered for the discharge, we further applied methods that consider information transmitted simultaneously (together) to the response variable. However, in this case, problems related to the redundancy of the predictors may appear, which is why we also applied concepts related to synergy and redundancy.

For multivariate analyses, we applied the following concepts:

The normalized multi-information (NMI) is defined by Ball et al., 2017 [5], as:(5)$$NMI\;(X_{1};X_{2};\ldots ;X_{n})=\frac{1}{n-1}\;TC\;(X_{1},X_{2},\ldots ,X_{n})$$
where TC is the total correlation.

Another measure of multivariate information is the redundancy–synergy index (RSI), which was created as an extension of the interaction information (Timme et al. 2014) [22]. If *S* is the set of predictor variables *X*_1_, *X*_2_, …, *X*_*n*_ and Y a predictand variable, then this index is defined according to the mutual information (MI) as:(6)$$RSI\;(S;Y)=\;MI(S;Y)-\sum_{X_{i}\in S}MI(X_{i};Y)$$

The redundancy–synergy index measures the interactions between a group of variables and another variable, except when *S* contains two variables, in which case the redundancy–synergy index is equal to the interaction information.

In the present investigation, before calculating the measures based on entropy, the time series were divided into three equally probable states. As shown in Foroozand et al. (2018) [8], choosing the number of bins must take into account the length of the data and, simultaneously, have sufficient detail to represent the distribution of the time series.

The results (Figure 2) obtained through applying Equation (5), considering five variables (four Palmer predictors and the response variable Q), must be discussed together with the results obtained by calculating the RSI (Equation (6)). Since we obtained few cases in which the four predictors had a positive RSI, we attempted to find the most suitable combination of the predictors by calculating the RSI for combinations of three and two predictors. The results with a positive RSI are presented in Appendix A.

The means of the NMI and RSI highlight the overall links between the predictor variables and predictand, but to obtain more detailed information, we applied wavelet transform, through which we obtained information on the behavior of the variables in the time–frequency domain.

Mutual information for multiple variables was acquired using the MATLAB routine [12].

#### 2.2.2. Multivariate Wavelet Approach

When analyzing a group of predictors for the estimation of a specific predictand, it is particularly important to outline the specific contribution of each one and to find a method to eliminate redundancy. In the following, we focus on solving this problem, which has been scarcely investigated to date.

An in-depth review of spectral methods and wavelet analysis, with applications for the climatic time series, is presented by Ghil et al. (2002) [64]. Wavelet transform is a universal mathematical method, which is able to work with nonstationary time series and is also able to detect when significant periods and their changes have been presented (Sleziak et al., 2015) [39].

As shown by Mihanović et al. (2009) [40], nonstationary wavelet analysis (Torrence and Compo, 1998) [34] was applied to obtain more detailed information on the time–frequency domain. In general, using wavelets, it is possible to analyze the time series according to a specific time scale.

A time series {X(t)} can be analyzed through its decomposition on several components according to two parameters: the *stretched/contraction* with s > 0 or s< 0, and a *translation* parameter u, −∞<u<∞. Such decomposition is performed through a real or complex function $\psi_{u,s}(t)$ called wavelet and defined as follows:(7)$$\psi_{u,s}(t)=\frac{1}{\sqrt{s}}\psi \left(\frac{t-u}{s}\right)$$

The continuous wavelet transform (CWT) of the time series X(t) is defined by:(8)$$W_{X}(u,s)=\int_{-\infty }^{+\infty }X(t)\frac{1}{\sqrt{s}}\psi^{*}\left(\frac{t-u}{s}\right)dt=\langle X(t),\psi_{u,s}(t)\rangle$$
analytic wavelet transform of X.

The * sign represents the complex conjugate of this expression.

An important role in wavelet transform is played by the mother wavelet ψ. It is necessary to mention the following properties (Sleziak, 2015, Mallat, 1998) [30,39]:


(a)The wavelet must have a mean value of 0:




$$\int_{-\infty }^{+\infty }\psi \left(t\right)dt=0$$




(b)The wavelet must have a final amount of energy:



$$\int_{-\infty }^{+\infty }{\left|\psi \left(t\right)\right|}^{2}dt=1$$



(c)It must have the condition of an inverse transform, where $\hat{\psi }(t)$ denotes the Fourier transform of the function ψ(t):




$$\int_{-\infty }^{+\infty }\frac{{\left|\hat{\psi }\left(\omega \right)\right|}^{2}}{\omega }d\omega <\infty$$



Cross-wavelet transform and the wavelet coherence provide information on the relationship between two time series. For the analysis of the covariance of two time series (Torrence and Compo, 1998) [34], the cross wavelet spectrum (XWT) of two time series, X and Y, with wavelet transforms W_X_ and W_Y_ is obtained as:(9)$$W_{XY}(s,u)=W_{X}(s,u)\;W_{Y}^{*}(s,u)$$
where the asterisk denotes complex conjugation, which can be considered as a measure of the correlation of the “wavelet spectra” of the two time series X(t) and Y(t). The cross-wavelet power, which is a measure of the common power, is calculated as $\left|W_{XY}\right|$.

A very useful tool is wavelet coherence (WTC). 
Coherence is a measure of the intensity of the covariance of the two series X 
and Y in the time–frequency space and, in accordance with Jevrejeva et al. (2003) 
[37], is defined as:(10)$$R^{2}(s,u)=\frac{{\left|\langle s^{-1}W_{XY}(s,u)\rangle \right|}^{2}}{\langle s^{-1}{\left|W_{XX}(s,u)\right|}^{2}\rangle \langle s^{-1}{\left|W_{YY}(s,u)\right|}^{2}\rangle }$$
where 〈⋅〉 is a suitable smoothing operator (Torrence and Webster, 1999) [35].

Sreedevi et al. (2022) [43] show that, because there is an interaction between the variables considered as predictors, a bivariate relationship can be clearly explained only by untangling the role of other contributory variables. This can be carried out by including only one predictand and one predictor in the analysis of the partial wavelet coherence (*pwc*), and by excluding the effect of the other predictor variables. The *pwc* method was suggested by Mihanović et al. (2009) [40] and then used in many studies, but in many of these studies, only 2 predictors were used, of which only the effect of one of the predictors was excluded.

Following (Mihanović et al. (2009) [40]), we present the wavelet coherence equations only for three variables. Let us consider time series {X1(t)}, {X2(t)} as predictors and {Y(t)} as the predictand.

According to Relation (10), without specifying the stretched/contraction and translation parameters, we have:(11)$${\left(R_{YX1}\right)}^{2}=R_{YX1}.R^{*}{{}_{YX1}}^{}\;{\left(R_{YX2}\right)}^{2}=R_{YX2}.R^{*}{}_{YX2}\;{\left(R_{X2X1}\right)}^{2}=R_{X2X1}.R^{*}{}_{X2X1}$$
and, the partial wavelet coherences squared (analog with first-order partial correlation (Legendre (2000) [65]) are defined as:(12)$${\left(RP_{YX1.X2}\right)}^{2}=\frac{{\left|R_{YX1}-R_{YX2}.R^{*}{}_{X2X1}\right|}^{2}}{\left(1-{\left(R_{YX2}\right)}^{2}\right)\left(1-{\left(R_{X2X1}\right)}^{2}\right)}\;{\left(RP_{YX2.X1}\right)}^{2}=\frac{{\left|R_{YX2}-R_{YX1}.R^{*}{}_{X2X1}\right|}^{2}}{\left(1-{\left(R_{YX1}\right)}^{2}\right)\left(1-{\left(R_{X2X1}\right)}^{2}\right)}$$

$RP_{YX1.X2}$ represents the partial coherence between predictand Y and predictor X1 after eliminating predictor X2.

The equations for several variables can be found in the work of Hu and Si (2021) [12], who generalize the partial coherence between the predictand and a set of predictors greater than 3.

Gu et al. (2022) [66] showed that Hu and Si (2021) [12] improved partial wavelet coherence (*pwc*) to effectively reveal the scale-dependent coherence, periodic characteristics, and lag relationship of two variables while avoiding the impacts of the other variables. Their improved partial wavelet coherence method has been successfully used in meteorology (Zhou et al. 2022) [67] and economics (Firouzi and Wang, 2021) [68], by untangling the scale-specific and localized relationships after excluding the effects of the other variables. In other words, *pwc* is appropriate for finding the partial correlation between response (predictand) variable Y and the set of predictor variables X, after eliminating the influence of the set of predictor variables Z.

Regarding the multiple wavelet coherence (*mwc*) developed by Hu and Si (2016) [3], the authors of this study applied it to find multiple coherence between different large-scale climate indices and the Danube discharge and between climate indices and indicators of solar/geomagnetic variability [18,19,20].

Confidence levels for multiple and partial coherence squared were calculated using the Monte Carlo method [38,69].

In the present investigation, as in cases where we had high redundancy (RSI < 0), we could not use all the predictors considered in this study together, we attempted to determine which details in the time–frequency domain are provided by partial wavelet coherence (*pwc*). As the highest redundancy (RSI < 0) was obtained by considering the 4 predictors in the summer season, we first analyzed this situation in accordance with Mares et al. (2022) [70]. The next negative RSI value (high in absolute value) was for the autumn season (Figure 2). Additionally, for this season, the highest correlation between the PC1 of the development in the MEOF of the 4 predictors and the Danube discharge was obtained (Mares et al., 2016) [44]. Therefore, the investigation based on wavelet analysis in the present study focused on the autumn season.

First, to verify the multiple coherence for the five variables, i.e., the 4 predictors (Palmer indices) together with the predictand Q, we calculated the multiple wavelet coherence (*mwc*) for the fall season using MATLAB routines provided by Hu and Si (2016) [3].

As shown by several authors [3,4,41,71,72], if the predictor variables are not independent, i.e., they are multicollinear, *mwc* can produce results with exceptionally high coherence, but this is unjustified from a statistical point of view. We obtained such a situation in the present study for the autumn season analyzed in this paper.

Therefore, in the next stage, we applied *pwc* between the Danube discharge and separately for each of the 4 predictors, excluding the other three predictors; that is, we applied *pwc* between the Danube discharge (Q) and each of the predictor variables (PDSI, PHDI, WPLM and ZIND), eliminating the other three variables. We performed all the experiments with the routine provided by Hu and Si (2021) [12]. For each case, the average of the coherence for the frequencies corresponding to the periods of 2–7, 8–15 and 22–30 years, and the boxplot statistics for each of the three periods bands were analyzed and discussed.

## 3. Results and Discussion

### 3.1. Testing the Linearity/Nonlinear Connections by Means of Mutual Information

As mentioned in Section 2.1., in accordance with the work of Mares et al. (2016) [44], we analyzed original Palmer indices in comparison with the Palmer self-calibrating indices. In their study, the analysis was performed simultaneously for each season, and the relationships were tested only using the Pearson correlation coefficient (R). For the overall analysis of MEOF using PC1, the closest link to the Danube discharge was obtained for fall, and separately, i.e., with each of the indices quantified by the PC1 of EOF decomposition, the results differed slightly, and maintained the closest connections during fall for PDSI and ZIND, while for PHDI and WPLM, the highest correlations with Q were in the summer season.

In the present study, first, we tested the nature of the relationship between the Palmer indices and the Danube discharge (Q), comparing the values of R with the nonlinear correlation coefficient NLR. Additionally, we analyzed the influence of the indices on the Q, both simultaneously and with a lag of 1 to 3 seasons before Q, for forecasting purposes.

In Appendix A, it can be seen that, in most cases, |R| > NLR, which leads us to the conclusion that the relations are linear. This is very clear for simultaneous connections in the same season.

In the following, we will underline the separate cases for each Palmer index, in which there are nonlinear links, i.e., NLR > |R|, but at the same time are significant; that is, in these cases, there is a higher predictability (Dionisio et al., 2006) [73]. We will also mention the cases in which there is no significant connection (linear or nonlinear) to the chosen CL of 99%.

(a)PDSI: there are two significant cases of nonlinearity: WIN with SPR (Q) and SPR with SUM (Q); an insignificant case from any point of view, i.e., PDSI in winter is not a good predictor for Q in FALL.(b)PHDI: there is only one case when NLR is almost significantly correlated with the chosen criterion (99%), and it is higher than |R|, namely, SPR with SUM (Q); there are two cases when this predictor has no significant connections with the predictor in any sense (linear or nonlinear), namely, WIN with FALL (Q) and SPR with FALL (Q).(c)WPLM: as in the case of PHDI, we can consider a nonlinearity for the combination of SPR with SUM (Q). There is only one situation in which there is no significant connection (linear or nonlinear) between WIN and FALL (Q).(d)ZIND: significant nonlinearity occurs in the case of the WIN predictor with SPR (Q). In two cases, there is no significant connection: ZIND from WIN with Q in SUM and FALL. Therefore, ZIND from winter does not provide any information for Q in summer and fall.

### 3.2. Analysis by Means of Multi-Information and Redundancy–Synergy Index

There are certain differences between the information (see Appendix A) transmitted separately by each of the four predictors to the predictand. Therefore, in the next stage, we tried to consider all four predictors, for which we calculated the total correlation and then the redundancy–synergy index.

The normalized multi-information (NMI) (Ball et al., 2017) [5] calculated according to Equation (5) is shown in Figure 2a.

From Figure 2a, it is obvious that the most information for the predictand is given by the four predictors during the summer season. However, is it correct to use all four predictors in a prediction relation (model or equation), assuming that we have identified them in climate simulation models or from previous influences of large-scale climate indices? It is not correct until we analyze the redundancy between them. In this sense, the redundancy–synergy index (RSI) is shown in Figure 2b [22]. As can be seen that the maximum summer NMI is associated with the highest RSI (in terms of absolute value). The higher the negative RSI value, the higher the predictor redundancy.

For the cases in which the RSI is positive (Figure 2b), (i.e., the predictors from winter with the predictand (Q) from summer and fall, as well as the predictors from spring with the Q from fall), we can consider all four predictors that provide significant information on Q’s behavior. Therefore, it was found that only for 3 out of 10 cases analyzed, the four predictors can be considered together in a mathematical relation to estimate the Danube discharge.

Taking into account the fact that only in three cases, we can consider the 4 predictors together to obtain information on the discharge behavior, in order to find other possible situations, we tested combinations of three and two predictors. For a single predictor, the results can be found in Appendix A. In Appendix A, only the results with a positive RSI are presented in the case of combinations of three or two predictors. We do not describe these cases here, because we will use them in a future paper in which we will develop regression equations for estimating discharge.

### 3.3. Applications of Partial Wavelet Coherence

Before obtaining the results by applying partial wavelet coherence (*pwc*), we focused on how coherent all five variables were (response variable Q and the four predictors) using multiple wavelet coherence (*mwc*).

Figure 3a shows *mwc* coherence for the fall season throughout the analyzed interval (1901–2000). As can be seen, the coherence of the five variables taken together has a high level of significance (>95%) over large areas of the cone of influence. Figure 3b shows the average coherence for frequencies corresponding to the periods of 2–7, 8–15 and 22–30 years. Regarding Figure 3c, it can be seen that the average of the period of 2–7 years has the largest extension in terms of the 75th percentiles of the sample.

The motivation for considering these period bands in the sample was to highlight possible influences on the interaction between the predictors and predictand. Interesting results were obtained in the work of Cai and Sakemoto (2022) [15], who applied *pwc* to determine the influence of El Niño on commodity prices. Authors analyzed time-varying wavelet coherence across different frequencies and found high correlations at lower frequencies (32–64 and 64–128 months), which occurred within two time intervals.

In the present study, the average coherences for the frequencies corresponding to the period of 2–7 years were chosen in order to highlight possible influences of large-scale climatic factors, such as quasi-biennial oscillations (QBO) [44,74], and other large-scale atmospheric factors, such as North Atlantic Oscillation (NAO) and Greenland-Balkan Oscillation (GBO) [18].

Additionally, as shown by Salinger (2005) [75], El Nińo Southern Oscillation (ENSO) is the primary global mode of natural climate variability in the 2–7-year time band. This was also recently emphasized by Malmgren et al. (2022) [76].

The previous investigations of the authors of this study (Mares et al., 2002) [74,77] revealed that the influence of NAO or ENSO in the Danube lower basin is not highly significant, as it is in other areas in Europe (Beranova and Huth, 2008 [78], (Bierknes and van Beek, 2009) [79] Lopez-Moreno et al., 2011) [80]. In the work of Rimbu et al. (2004) [81], it is shown that the physical mechanisms behind NAO–ENSO interactions and their variable impact on the European climate are unclear.

The last two averages for the periods between 8–15 and 22–30 years, respectively, can highlight the influence of the Schwabe and Hale solar cycles [18,19,20,44,82,83,84,85,86].

In accordance with the results obtained in the work of Mares et al., 2016 [44], we present the power spectra of the Danube discharge and of PC1-MEOF in the autumn season in Appendix A. From Appendix A, we observe periodicities with certain levels of confidence, which are related to internal atmospheric variability or with external factors such as solar activity.

Interesting results were obtained by Neyestani et al., 2022 [87], where different periodicities are highlighted, by applying wavelet coherence between the temperature, precipitation and climatic signals for some zones over Iran in different frequency bands. The findings are related to intra-annual, annual, inter-annual and decadal oscillations.

Zamrane et al. [88] also applied wavelet analysis, in which certain modes of climate variability between rainfall and runoff for northwest Africa area were found.

Returning to the values of multiple coherence in Figure 3, all three panels (Figure 3a,b,c) indicate particularly high mean values of coherence, which are due to the redundancy of the predictors, as also shown by the analysis of TC or NMI (Figure 2a). This is why it is necessary to eliminate this redundancy, which we achieved by applying partial wavelet coherence (*pwc*).

Thus, for the fall season for which we performed the analysis in the present study, we applied *pwc* between the Danube discharge and separately for each of the four predictors, excluding the other three predictors. In Figure 4, Figure 5, Figure 6 and Figure 7, the results of the *pwc* analysis between the Danube discharge (Q) and each of the predictor variables (PDSI, PHDI, WPLM and ZIND), eliminating the other three variables, are presented. In these figures, in the first panel (a), *pwc* depends on the period and years considered (1901–2000); in the second panel (b), the average coherence for the frequencies corresponding to the periods of 2–7, 8–15 and 22–30 years is shown; and in the last panel (c), the boxplot statistics for each of the three periods bands analyzed in (b) are presented. From the analysis of the four figures, it is observed that each case has certain particularities; the greater coherence depends on a certain frequency (here, the periods are expressed in years) and on a certain time interval.

However, we attempt to find a correlation between the four cases analyzed. Thus, if we refer to the band of the period of 2–7 years, it presents (Figure 4, Figure 5 and Figure 6; that is, *pwc* between Q and PDSI, or PHDI, or WPLM, by eliminating the other three corresponding variables) the greatest coherence around the 1970s for the first three cases, but this coherence does not extend over a long period of time. This can be seen in panels (c) corresponding to these situations, where 75% of cases do not exceed 0.7 or slightly exceed 0.7 in the case of Q of PHDI. Regarding Q with ZIND, the mean coherence for the period of 2–7 years had a maximum of very short intervals of time towards the end of the 20th century (Figure 7a,b).

Related to the second band of the period of 8–15 years, it was observed that the highest coherence was obtained for *pwc* of Q with PHDI (Figure 5a–c) and Q with ZIND (Figure 6a–c), which were located towards the end of the analyzed time interval.

Concerning the band of the period of 22–30 years, it was observed that the evolution of the coherence of *pwc*, except for the situation when WPLM was considered as a predictor (Figure 6) for the Danube discharge (Q), showed significantly high values for relatively long intervals (approximately 40 years from 1960 to 2000). Additionally, the statistics presented in Figure 4c, Figure 5c and Figure 7c indicate the same findings. It should be noted that the most obvious case of these three situations is *pwc* between Q and ZIND, where the coherence is the highest for the period of 22–30 years compared to the other two periods.

Although the physical interpretation is not easy to achieve, we try to speculate that where there is significant coherence between the discharge in the lower Danube River basin and Palmer indices, which were considered simultaneously in this study, there are common causes that can be associated with different types of interactions and influences. Thus, if, for the period band of 2–7 years, we can consider that there are influential internal mechanisms of the atmosphere, or of the complex ocean–atmosphere interactions (Su et al., 2019) [71], for the frequencies corresponding to the periods of 8–15 and 22–30 years, this common coherence may be due to extra-atmospheric factors, such as solar/geomagnetic variability.

In the present study, the results related to oscillations are consistent, for the most part, with the periodicities found in the paper [89] (Briciu and Mihaila, 2014). In the mentioned work, for South-East Europe, the influence of some climatic oscillations and sunspot number on river discharge in Romania, Ukraine, and Moldova was analyzed, by applying wavelet transform. It was found that the rivers have periods with a frequency of 2.9–3.5, 8.3, 11.7, and 27.8–33.1 years, which can be associated as the authors points out, with periodicities of precipitation, climatic indices and solar activities.

Certain of our findings related to the common cyclicities between Q and Palmer indices for certain time intervals are somehow in agreement for the 8–15 year periods, with those obtained by Szolgayova et al. (2014) [90]. Thus, in Figure 3b, the coherence (*mwc*) obtained between the discharge of the Danube and the four Palmer indices, for the averaged periods 8–15, is the highest between the years 1930–1960. In the work [90], where four stations along the Danube were analyzed, cycles with 11–15-year periods, both in discharge and precipitation were found between 1935 and 1975.

However, which procedure should be adopted to select predictors? For cases combining predictors (of four, three or two variables) that are not redundant, we can use regression equations. If there is high redundancy, as in the case of the autumn season, we can choose one of the cases analyzed here via *pwc*, depending on the predictor’s response to external causes or large-scale atmospheric factors, such as NAO and especially GBO for the lower Danube basin. The influence of the NAO on precipitation and river discharge for the western European region, has been well documented in many publications, as shown in the recent work Lorenzo-Lacruz et al. (2022) [91].

This is due to the fact that it is more difficult to determine the response of the Danube discharge to these factors, while Palmer predictors, which are generally calculated based on temperature, precipitation and soil conditions, respond better to extrapolation methods.

For a clearer view of the contribution of each of the three period bands, in Figure 8a,b,c, the overlapping coherence for the four *pwcs* is analyzed. For example, in Figure 8c, where the average coherence for the band of 22–30 years is provided, the highest coherence is evident in the case of maintaining ZIND with Q, and eliminating the other three predictor variables.

Therefore, this study constitutes a step in our investigations of the discharge of the Danube River in the lower basin using different methods, and considering some of the many possible predictors.

In our previous investigations [18,19,20,44], as in the present study, we separately considered the influence of atmospheric predictors, such as Palmer indices, large-scale climate indices, and indicators of solar/geomagnetic activity, regarding their information content, on the Danube discharge.

In the future, we intend to analyze a combination of different atmospheric and extra-atmospheric predictors in order to select the most significant ones for estimating the Danube discharge, taking into account both the synergy and the redundancy of the predictor–predictand relationship.

Predictors can be obtained from meteorological and hydrological data through simple extrapolations if they have linear structures, or using an artificial neural network method, as shown in the work of Elbeltagi et al., 2022 [92], for nonlinear structures.

## 4. Conclusions

From the linear (R) and nonlinear (NLR) correlations, it can be concluded that the simultaneous connections in the same season between the Danube discharge (Q) in the lower basin and the Palmer indices (PI) in the upper and middle basin are linear.

In terms of simultaneous links in the redundancy–synergy index (RSI) analysis, for the Q estimate in the autumn season, we did not find any combination of predictors (four, three or two) that were synergistic and nonredundant.

For different lags, few cases were obtained in which all four predictors could be considered together to provide a significant information content for the discharge evolution. These situations are: predictors (Palmer indices) from winter with Q in summer and autumn and predictors from spring with Q in autumn. Additionally, several combinations of three or two predictors from winter and spring were found that could be considered significant (high NMI and positive RSI) for estimating autumn Q.

These findings lead to the use of these combinations of predictors to estimate the discharge in autumn. Therefore, these combinations meet the conditions to be considered in a robust regression equation.

The results of the NMI and RSI highlight the overall links between the predictor variables and predictand, but to obtain in-depth information, we applied wavelet transform, through which we obtained information on the behavior of the variables in the time–frequency domain.

In the multivariate wavelet analysis, through the application of *pwc*, we obtained certain information on the interaction between Q and each of the four Palmer indices, eliminating the contribution of the other three, which is detailed in the frequency–time domain.

For each individual case, we presented the distribution of coherence in the frequency–time domain; the average coherence for the frequencies corresponding to the periods of 2–7, 8–15 and 22–30 years; and the boxplot statistics the three periods considered.

The most significant coherences between the discharge and the Palmer indices during the fall season were found between Q and the Palmer ZIND index in the second half of the 20th century. Regarding these findings, it can be speculated that they are related to the influence of the variability of the double solar cycle. The other coherences for smaller periods may be associated with the variability of climate indices on a regional or global scale.

In light of these findings, we will continue our investigations into the atmospheric and extra-atmospheric causes of Q variability in the lower Danube River basin, considering several predictors, and based on both information theory and multivariate wavelet transform measures. The results obtained in the present study are useful for estimating the discharge, as Palmer indices can be estimated from the simulated data by applying General Circulation Models or Regional Climate Models. These new findings will help to improve our knowledge of the predictability of the Danube discharge.

## Figures and Tables

**Figure 1 entropy-24-01375-f001:**
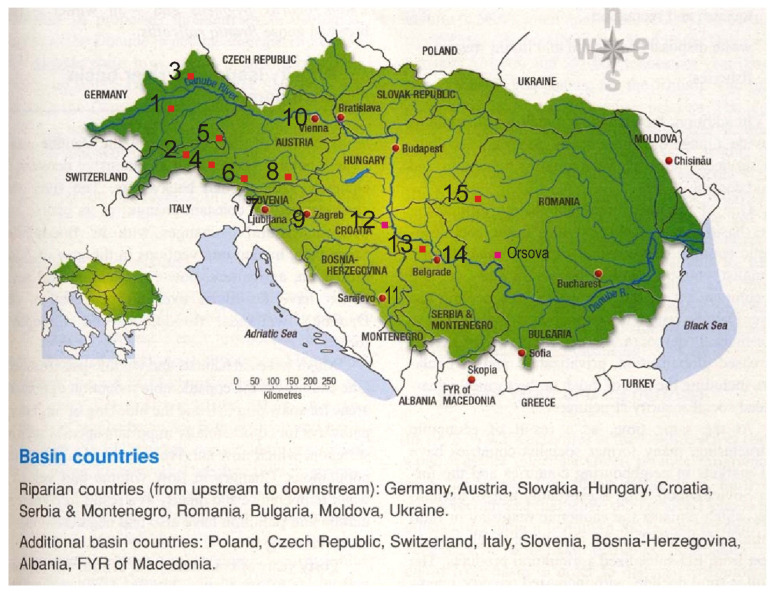
Localization of 15 stations situated upstream of Orsova hydrological station.

**Figure 2 entropy-24-01375-f002:**
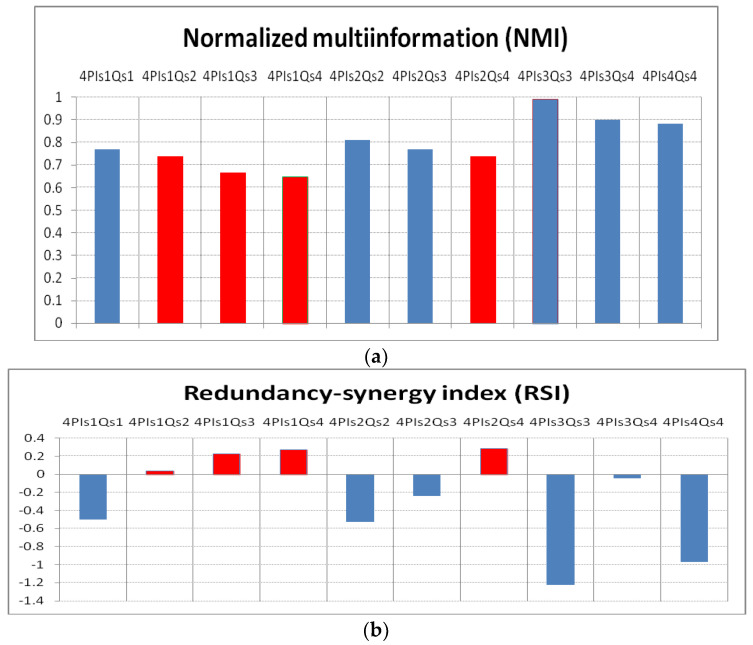
(**a**) The normalized multi-information (NMI), for the 10 combinations of predictors with the predictand; (**b**) redundancy-synergy index (RSI) for the same combinations. With red in Figure 1a are the corresponding positive RSI values.

**Figure 3 entropy-24-01375-f003:**
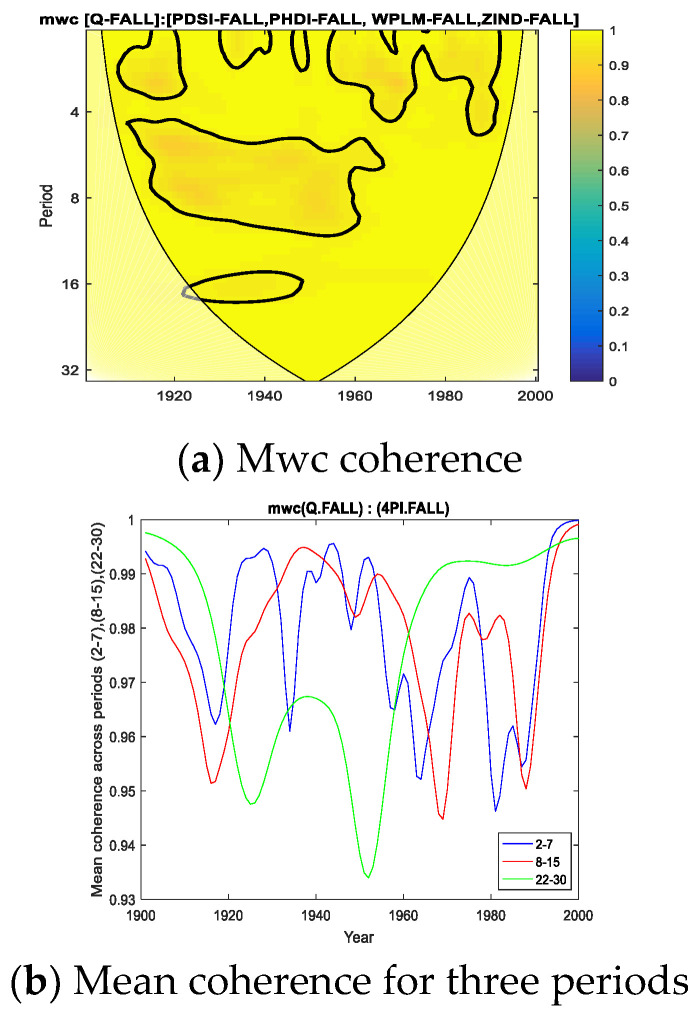
*Mwc* for Palmer indices and Danube discharge for FALL (1901–2000).

**Figure 4 entropy-24-01375-f004:**
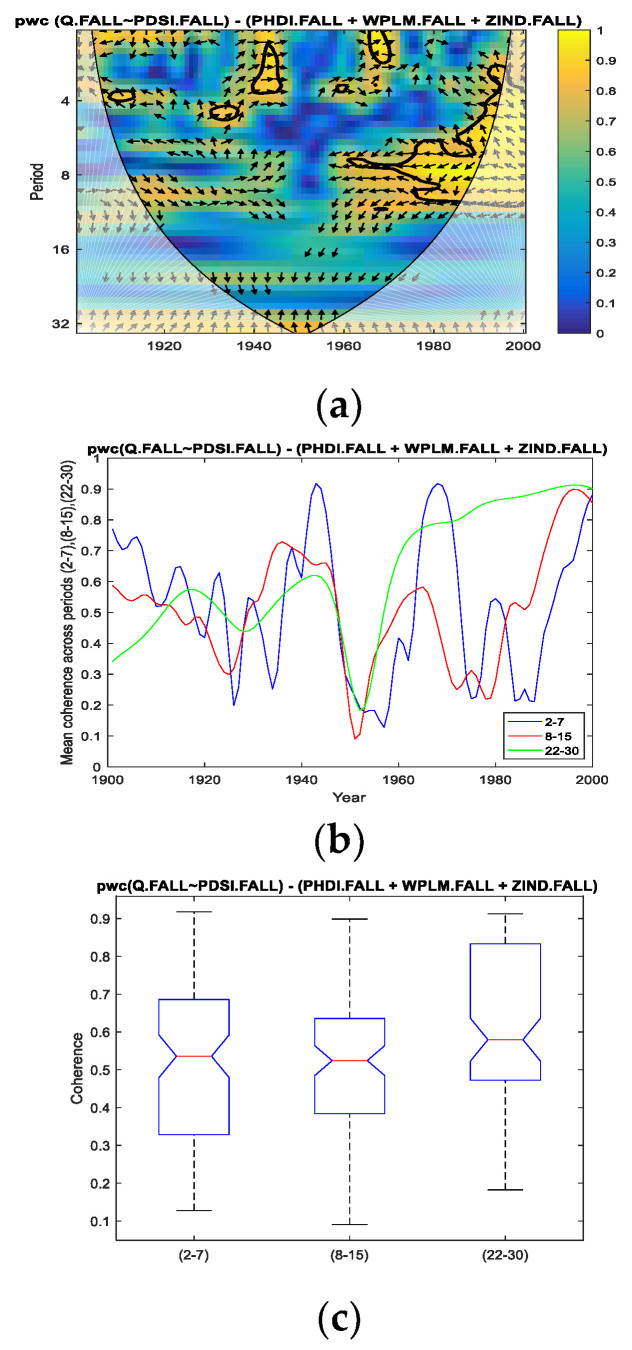
*pwc* for Q with PDSI, excluding PHDI, WPLM and ZIND. (**a**) frequency-time distribution of coherence; (**b**) average coherence across the frequencies corresponding the periods (2–7), (8–15), (22–30); (**c**) boxplot corresponding to the periods in (**b**).

**Figure 5 entropy-24-01375-f005:**
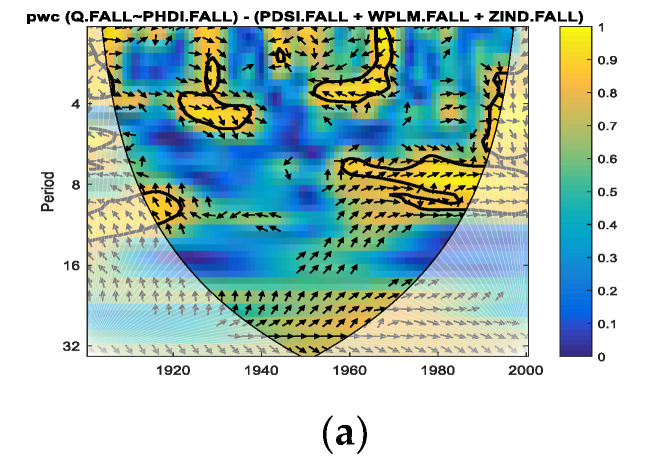
*pwc* for Q with PHDI, excluding PDSI, WPLM and ZIND. (**a**) frequency-time distribution of coherence; (**b**) average coherence across the frequencies corresponding the periods (2–7), (8–15), (22–30); (**c**) boxplot corresponding to the periods in (**b**).

**Figure 6 entropy-24-01375-f006:**
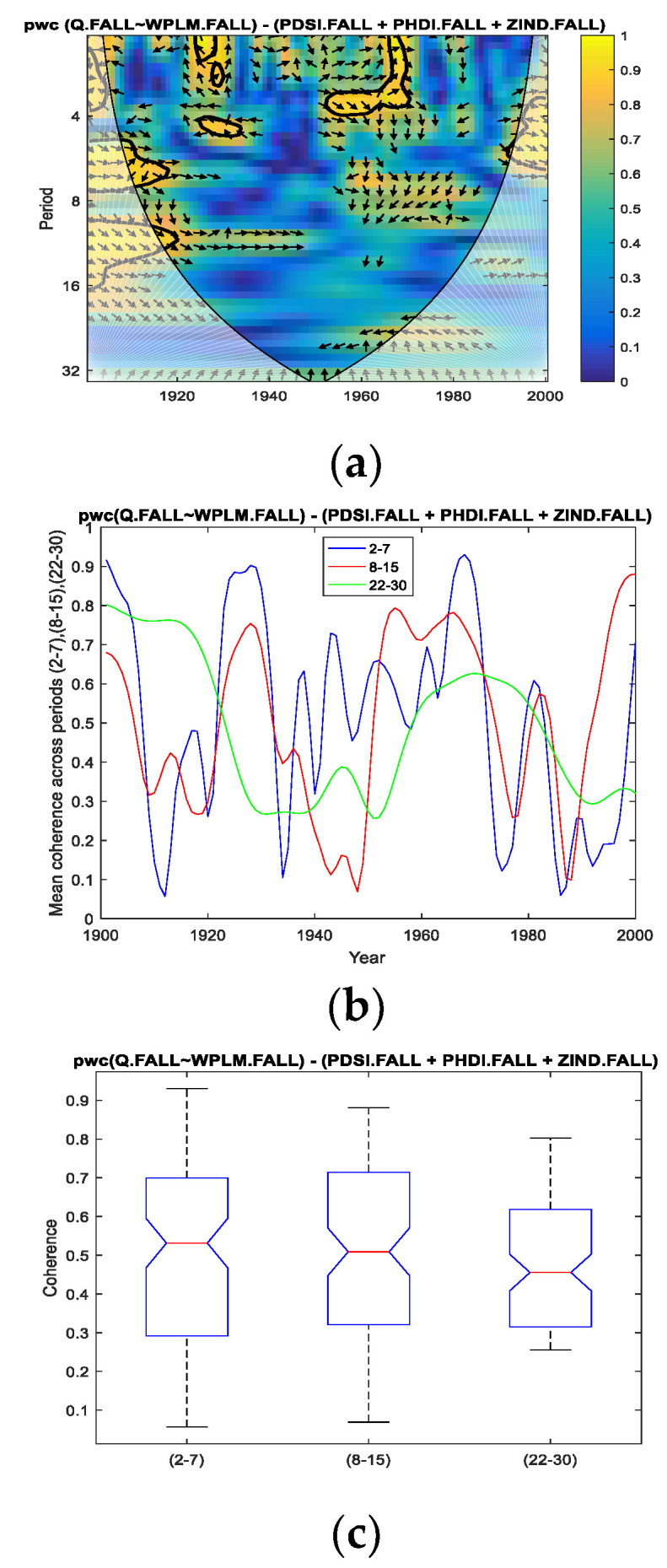
*pwc* for Q with WPLM, excluding PDSI, PHDI and ZIND. (**a**) frequency-time distribution of coherence; (**b**) average coherence across the frequencies corresponding the periods (2–7), (8–15), (22–30); (**c**) boxplot corresponding to the periods in (**b**).

**Figure 7 entropy-24-01375-f007:**
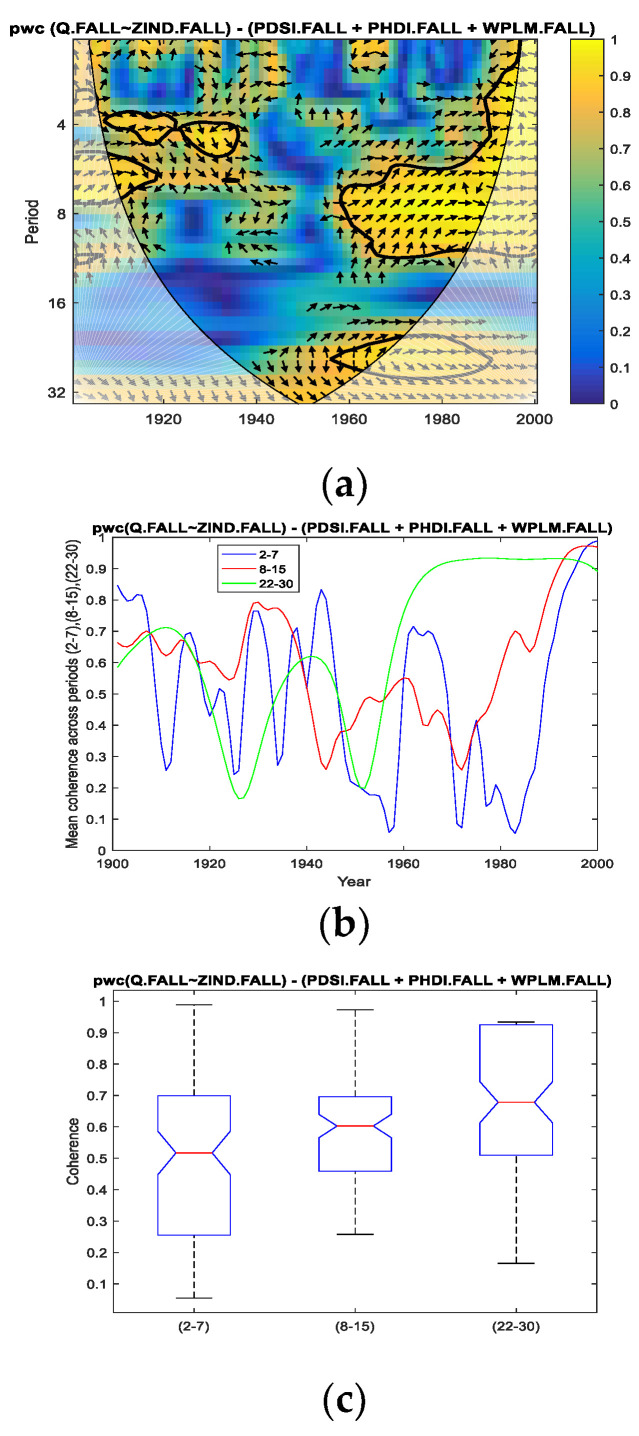
*pwc* for Q with ZIND, excluding PDSI, PHDI and WPLM. (**a**) frequency-time distribution of coherence; (**b**) average coherence across the frequencies corresponding the periods (2–7), (8–15), (22–30); (**c**) boxplot corresponding to the periods in (**b**).

**Figure 8 entropy-24-01375-f008:**
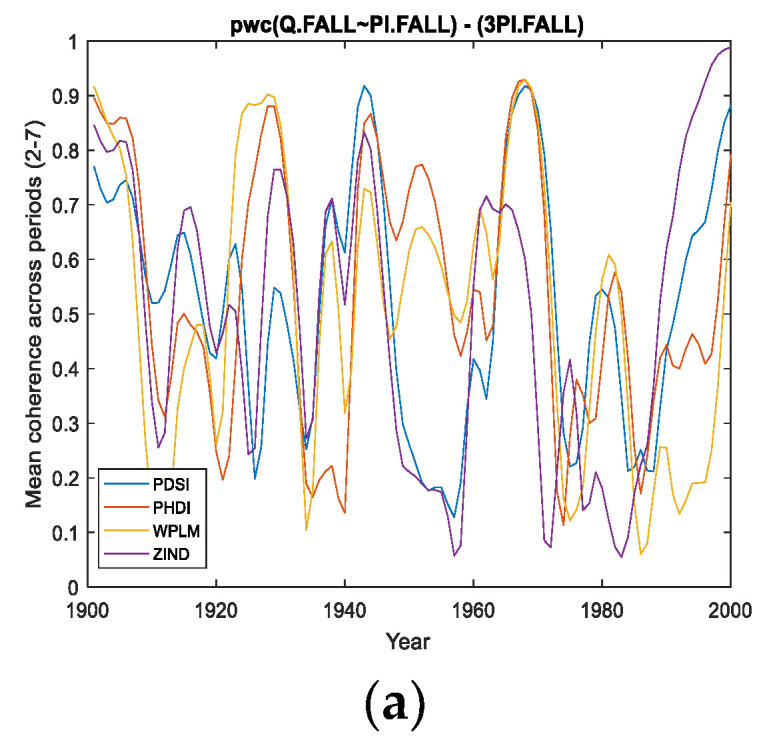
Average Coherence across the frequencies corresponding the periods (**a**): (2–7); (**b**): (8–15); (**c**): (22–30). PDSI means *pwc* [Q with PDSI- (PHDI, WPLM, ZIND)]. Same with the other indices.

**Table 1 entropy-24-01375-t001:** Geographical information about the analyzed stations in different. countries (CN) with the Available Water Capacity (AWC).

Station	CN	LONG	LAT	Height	AWC (mm)
Augsburg	GE	10.56	48.26	463	100
Innsbruck	AT	11.24	47.16	577	15
Regensburg	GE	12.06	49.02	365	100
Sonnblick	AT	12.57	47.03	3106	15
Salzburg	AT	13.00	47.48	437	15
Kredarica	SI	13.51	46.22	2514	50
Ljubljana	SI	14.31	46.04	299	15
Graz	AT	15.27	47.05	366	15
Zagreb	HR	15.58	45.49	156	150
Wien	AT	16.21	48.14	198	15
Sarajevo	BA	18.23	43.51	577	50
Osijek	HR	18.38	45.32	88	150
Novi-Sad	RS	19.51	45.20	84	150
Beograd	RS	20.28	44.48	132	150
Arad	RO	21.21	46.08	117	150

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
