# Peer review of "Selection of Optimal Palmer Predictors for Increasing the Predictability of the Danube Discharge: New Findings Based on Information Theory and Partial Wavelet Coherence Analysis"

_entropy, 2022, doi:10.3390/e24101375_

Round 1

Reviewer 1 Report

The paper describes a methodology for selecting  predictors to estimate seasonal discharge in the lower Danube basin based on information theory and wavelet coherence. The article is overall interesting since it combines different data oriented methods to achieve the goal of estimating seasonal discharge. Although the paper uses information theoretic methodologies that fits the scope of the entropy journal, some major revisions shoul be performed in order for the paper to be publishable. In the following, I recomend the authors correct the paper according to the following major and minor adjustments:

Some major adjustements that improve the overall paper are:

The abstract should transmit the main result of the paper, however, in my opinion, the results are not clearly stated and the authors should include it.

The authos shoud make a major revision of the grammar and the way the citations are made. In this context, citations should be homogeneous and some grammatical errors made the paper difficult to follow. In addition, authors should avoid long sentences that make some paragraphs difficult to read. It is preferred to have many short paragraphs than a single long one.

The information theory subsection is clearly reviewed, however, the multivariate wavelet approach is vaguely reviewed and no concept or equation is included. I believe that by the inclusion of wavelet-related concepts and equations, the paper will be more complete and self-contained.

In line 95, tha author claim that the procedures for applying the wavelet transfrom are briefly described, however, in my point of view there is not such a description.

The methods sections should describe the way in which experiments will be carried out. The authors should include this and not describe briefly in each subsection of section 3 (results and discussion), the way in which some methodologies are applied.

Figures 2-7 are nor described in detail, the authors should provide a standard description for each figure and increase the font-size of the y-axis,  x-axis labels and legends. Moreover,  the abstract and title claims that the predictor selection will allow to estimate (predicts) seasonal discharge, however, the paper does not provide an estimation.

The word "in view" in the title can be optionally omitted.

Some minor adjustments in order for the paper to be more readable are the following:

In the abstract, please provide the meaning of EOF (in this case, empitical orthogonal functions (EOF)). In addition, it is common practice to write the abstract in a continous way with no line breaks in order for the abstract to look like a single long paragraph.

The authors make use of the word "informativeness" within the paper, however, a more comprehensive and standard way is "information content". In addition, within the same paagraph wavelet transform for multivariables shoud be written multivariate wavelet transform.

Please provide an explanation of the terms of equation (5), e.g., where TC is the....

Informativity in section 3.2 may be changed to "information". 

Reviewer 2 Report

Dear,

The study assesses statistical predictors and methods, based on the Palmer drought index, to predict the seasonal Danube discharge. The approach proposed is relevant to climate, and hydrology areas and the methods are adequate, with emphasis on the use of wavelets. However, for publication, the article needs to improve several sections.

# Minor review

  • The abstract must be written in continuous text in a single paragraph;
  • Define the acronyms PC1 - Principal Components 1 and EOF - Empirical Orthogonal Function;
  • Move Fig.1 to the final of the manuscript;
  • The location of the figures in the text must be indicated;
  • Improve de Figure quality. See other papers of Entropy;
    • Provide a map with weather and hydrological stations, political boundaries, river basin, hypsometry, and other elements that facilitate locating the study area.

# Major review

  • The Material e methods need to be improved:
    • Improve the description of databases and series. what is the size of the series? Start and end of series? Were the series submitted to data quality analysis? Were the series continuous? If not, gap filling was performed. By what method? What about the homogeneity analysis?
    • It is necessary to present Eqs. Basics and details of Palmer's indices used;
  • There is a need to improve the discussion, based on climate variability (ENOS and others) and associated climatic and hydrological processes. Most of the results are descriptive and only evaluation of the methods and indices.

Best regards

Round 2

Reviewer 1 Report

The revised version of the paper contains all the suggestions provided by tha reviewer. I consider the paper more readable, self-contained and is now more publishable than the first version submitted to the journal.

Author Response

We are grateful for your very useful suggestions and comments, that helped us improve our manuscript.

Reviewer 2 Report

Dear,

The Authors need to be improve the tables and figures quality, main the Fig. 1 (map of study area)

Improve the discussion in relation to results of other studies in the Europe.

Author Response

Thank you very much for your useful suggestions and comments that helped us improve our manuscript.

In the new version, we have replaced Fig.1 with another one, that we believe is suitable for readers.

We have also improved the quality of Table 1.

We developed the discussions related to the results obtained in other studies for the areas of Europe.

Please find these new discussions introduced on lines 539-552 and 557-559, and the corresponding new references can be found on numbers 89, 90 and 91.

All changes in the new version of the manuscript are colored in red.
